# Revisiting Coarse-to-fine Paradigm in Nighttime Flare Removal via Visual Prompt

## Abstract

Flare removal is a crucial task in image processing, aiming to eliminate unwanted lens flare. Existing end-to-end flare removal methods, despite their progress, often introduce artifacts in the restored images. While multi-stage (coarse-to-fine) strategies in image restoration have proven effective for artifact suppression, their direct application to flare removal tasks yields limited improvements, raising questions about their inherent suitability. Inspired by inpainting techniques and prompt-based learning, we propose a plug-and-play Prompt Inpainting Network (PIN) that redefines coarse-to-fine processing for flare removal. We define this process as a Prompt Inpainting Pipeline (PIP). Our PIP introduces two synergistic mechanisms: Firstly, it leverages predicted flare mask from the coarse flare removal stage to explicitly exclude corrupted pixels and guide context-aware restoration. Second, high-quality decoder features from the coarse stage are re-purposed as visual prompts to condition the refinement network, enabling feature-aware structural consistency in refinement stage. PIP is designed as a model-agnostic pipeline that seamlessly integrates with arbitrary restoration architectures, while introducing negligible computational overhead (minimum 1% parameters increment). Experiments demonstrate that PIP significantly reduces artifacts and achieves state-of-the-art performance across multiple benchmarks, proving that coarse-to-fine paradigms-when augmented with explicit corruption exclusion and visual prompts-are indeed effective for flare removal.

## 1 Introduction

Flare in images is an optical phenomenon caused by the complex structure of imaging systems and lens contamination or wear. When light enters the camera, it can be reflected or scattered along unintended paths, leading to the formation of flare. This results in the appearance of bright, often streaky or shimmering regions in the image, which not only degrades the overall visual quality but also poses significant challenges to downstream visual tasks (*e.g.*, semantic segmentation, depth estimation) Dai et al. (2023a). Therefore, how to remove flare in images has attracted plenty of attention from industry and academia Asha et al. (2019); Vitoria & Ballester (2019); Feng et al. (2023); Sun et al. (2020); Wu et al. (2021); Dai et al. (2022). Wu et al. Wu et al. (2021), Dai et al. Dai et al. (2022), and Dai et al. Dai et al. (2023b) propose the dataset for flare removal and adopt existing end-to-end image restoration networks Zamir et al. (2021); Chen et al. (2021); Wang et al. (2022); Zamir et al. (2022) to learn complex patterns in flare-corrupted images and generate corresponding flare-free versions. However, they are plagued by a persistent problem: artifacts are produced in the restored images. As shown in Figure 1, when employing Flare7k++ pipeline Dai et al. (2023a) to remove flare from the flare-corrupted image, the restored images exhibit noticeable artifacts.

The image restoration community has long embraced coarse-to-fine pipeline to address similar challenges Zamir et al. (2021); Chen et al. (2021); Li et al. (2023); Nehete et al. (2024). These methods typically consist of a coarse-level restoration stage, where the major defects or degradations are roughly corrected, followed by a refinement stage that focuses on enhancing the details and removing any remaining artifacts. However, when directly applying this simple coarse-to-fine strategy to the flare removal task, it has shown limited effectiveness. The comparison results illustrated in Figure 1 (b) and (c) reveal critical insights into the limitations of simple coarse-to-fine paradigms in flare removal tasks. Figure 1 (b) evaluates single-stage (UNet Ronneberger et al.

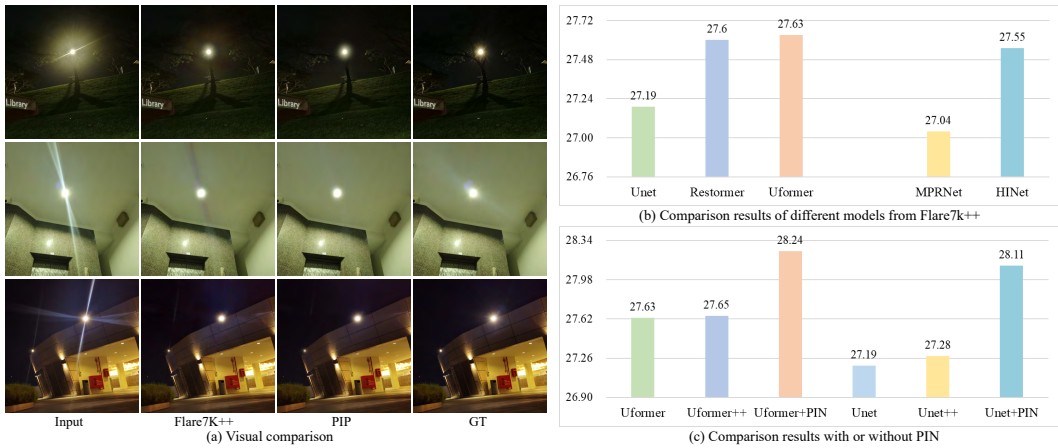

Figure 1: (a) Visual comparison of flare removal on real-world nighttime flare images and we adopt Uformer as the base network. Based on the comparison, our PIP pipeline provides significant improvement. (b) Comparison results of different models from Flare7k++. (c) Comparison results with or without PIN.

(2015), Uformer Wang et al. (2022), Restormer Zamir et al. (2022)) and multi-stage (MPRNet Zamir et al. (2021), HINet Chen et al. (2021)) methods on the Flare7K++ benchmark Dai et al. (2023a). Surprisingly, the multi-stage methods do not outperform single-stage methods. MPRNet, with more parameters, even lags behind the simple Unet. Figure 1 (c) further investigates this phenomenon by appending a simple U-shape refinement network to pretrained Uformer and Unet from Flare7K++ Dai et al. (2023a), denoted as Uformer++ and Unet++. However, these enhanced models exhibit negligible performance gains over their single-stage counterparts, indicating the ineffectiveness of the simple coarse-to-fine paradigms in flare removal.

This raises a question: Is the coarse-to-fine approach unsuitable for flare removal, or can it be optimized? Inspired by inpainting techniques Dong et al. (2022); Podell et al. (2023); Suvorov et al. (2022) and concept of prompts Brown et al. (2020); Rombach et al. (2022); Zhang et al. (2023b), we propose a plug-and-play Prompt Inpainting Network (PIN) that redefines coarse-to-fine processing for flare removal. We define this process as a Prompt Inpainting Pipeline (PIP). Similar to the previous methods, we separate the flare removal process into a coarse flare removal stage and an image refinement stage. Since our PIP is designed as model-agnostic pipeline, we can use any U-shape networks to coarsely generate the flare-removed image and the flare in the coarse flare removal stage. The predicted flare image is then used to create a mask (via our Mask Block), which precisely identifies the areas in the image that are contaminated by flare. By masking out these regions, we can isolate the problem areas and focus on restoring them. Subsequently, we use pixels from the non-polluted regions with better quality to rewrite the details in the masked areas. These pixels carry more accurate color, texture, and semantic information, which can help to restore the flare-affected areas more realistically. In addition, we propose prompt calibration block which adopts high-quality features extracted from the coarse flare removal stage as a visual prompt to guide the rewriting process and generate promising flare-free and artifact-free images. In this fashion, our proposed PIP pipeline effectively reduces the occurrence of artifacts while removing the flare, as show in Figure 1. We conduct extensive experiments to evaluate the effectiveness of PIP across diverse architectures, ranging from lightweight CNN-based models like Unet Ronneberger et al. (2015) to heavyweight Transformer-based designs such as Uformer Wang et al. (2022) and FF-Former Zhang et al. (2023a). In all cases, our PIP achieves significant performance improvements.

Our contributions can be summarized as follows:

- We redefine coarse-to-fine processing for flare removal and propose the Prompt Inpainting Pipeline (PIP). By leveraging inpainting concepts and visual prompts effectively addressing the ineffectiveness of traditional coarse-to-fine methods in flare removal.

- Our Prompt Inpainting Network (PIN) is model-agnostic and can be seamlessly integrated into any existing flare removal network, whether based on CNN or Transformer.

- Extensive experiments across diverse architectures demonstrate that our PIP can achieve substantial performance improvements in flare removal.

## 2 RELATED WORK

### 2.1 DATASETS

Aiming for a low-level task, the performance of flare removal methods depends on the qualities of the datasets. A dataset that contains a large amount of data pairs can greatly improve the performance of neural networks when handling real-world scenarios. To this end, Wu et al. Wu et al. (2021) propose a semi-synthetic dataset that contains 2001 captured flare images and 3000 simulated flare images. However, they focus on daylight flare and the proposed dataset tends to be less generalization to real-world flare which can be captured by diverse lenses and light sources, especially at nighttime. Therefore, Dai et al. Dai et al. (2022) propose a nighttime flare removal dataset Flare7K, which contains 5000 scattering and 2000 reflective flare images. Furthermore, Dai et al. Dai et al. (2023a) propose Flare7K++, which is an extended version of Flare7K. On the basis of Flare7K, Flare7K++ gives each image an additional light source annotation and proposes a new real-captured subset Flare-R which contains 962 flare images. These datasets exhibit high sensitivity to scattering flares while giving insufficient attention to reflective flares. Hence, Dai et al. Dai et al. (2023b) propose the first reflective flare removal dataset named BracketFlare dataset based on the prior that the reflective flare and light source are always symmetrical around the lens's optical center. They employ continuous bracketing to capture the reflective flare pattern in unexposed images and aggregate with exposed images to synthesize paired data. They conduct experiments and prove that neural networks trained on this dataset gain the capability of removing the ghosting effect in images.

### 2.2 NETWORK STRUCTURE

As capturing large amounts of data pairs for flare removal is challenging and tedious, earlier deep learning methods He et al. He et al. (2010) tend to utilize unsupervised methods. Qiao et al. Qiao et al. (2021) propose a generative adversarial network-based learning framework to learn from unpaired data. They adopt the idea of cyclegan Zhu et al. (2017) and separately detect the light source region and the flare region. The output is generated by blending the flare-removed image and the detected light source mask. Their method achieves promising results when handling tiny light sources and flares, whereas fail on images with strong light sources and large flares.

As multiple synthetic and real flare removal datasets have been proposed, Wu et al. Wu et al. (2021) and Dai et al. Dai et al. (2022) adopt many end-to-end image restoration methods Ronneberger et al. (2015); Chen et al. (2021); Zamir et al. (2022); Wang et al. (2022) to extract the flare image from the input flare-corrupted image. They surpass the unsupervised methods, whereas still generate artifacts during the flare removal process as they view the gap between the flare-corrupted image and the flare-free image equal to the flare image. Meanwhile, Dai et al. Dai et al. (2023a) propose a different pipeline named flare7k++, which separately extracts the flare and restores the image in a simultaneous manner and adopts image restoration networks to do both jobs instead of simply generating the flare-free image. However, such a method compels the network to analyze the flare pattern and generate flare-free images simultaneously, which some artifacts still happen, especially when a strong streak appears. To this end, we propose a model-agnostic PIP pipeline to reduce the flare and generate images with fewer artifacts. By separating the flare removal process into a coarse flare removal stage and an image refinement stage, our pipeline surpasses state-of-the-art methods.

## 3 METHODOLOGY

To better illustrate our PIP pipeline and distinguish it from the previous pipeline, we introduce the Flare7k++ pipeline first. Sequentially, we illustrate the details of our PIP pipeline and demonstrate its superiority.

### 3.1 FLARE7K++ PIPELINE

As shown in Figure 2 (a), the Flare7k++ pipeline adopts a network with a U-net backbone to predict the flare-free image and the flare image which excludes the light source information. In this way, the network can preserve the light source image and better locate the flare image by individually estimating the flare image. However, due to the pixel limit of digital systems, some information is

blocked by overexposed streaks. Such degradation is hard to compensate for as the blocked area contains little useful information and the Flare7k++ pipeline requires the network to extract the flare and generate flare-free images simultaneously, thereby giving too much burden for the neural network and leading to severe artifacts. To this end, we propose our two-stage pipeline named PIP to reduce the artifacts.

### 3.2 PROMPT INPAINTING PIPELINE (OURS)

Our PIP pipeline provides a novel perspective for the flare removal task. Existing flare removal pipelines accomplish this task under the idea of image-to-image translation. However, such a design may be suboptimal for the flare removal task as the streak and shimmer in the flare-corrupted image occasionally appear to be strong and fully block the content embedded and these methods may generate severe artifacts when handling such flare. Therefore, we propose the PIP pipeline, which accomplishes this mission by rewriting the image details occupied by the flare.

Figure 3 shows the details of the PIP pipeline. As a coarse-to-fine two-stage pipeline, the PIP pipeline consists of a coarse flare removal stage and an image refinement stage. Concretely, our PIP pipeline is a model-agnostic pipeline. During the coarse flare removal stage, any U-net flare removal method can be employed as

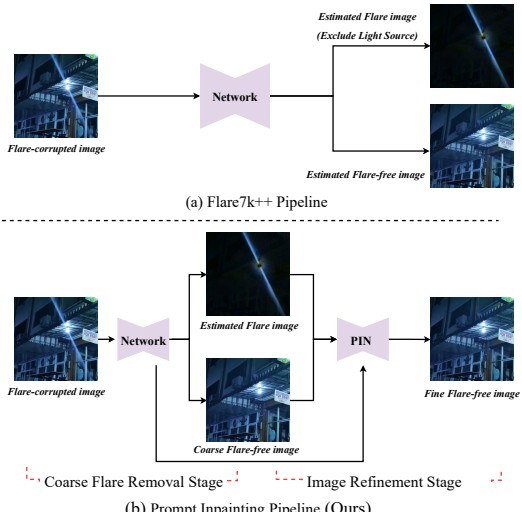

(a) Flare7k++ Pipeline

(b) Prompt Inpainting Pipeline (Ours)

Figure 2: Comparison between Flare7k++ pipeline Dai et al. (2023a) and our PIP pipeline.

multi-scale features are useful in such image translation tasks, and most methods in this task adopt one U-net structure as the backbone to extract multi-scale features. As for the image refinement stage, we propose the Prompt Inpainting Network (PIN) and employ a U-net backbone under the guidance of the multi-scale features extracted during the first stage.

### 3.2.1 COARSE FLARE REMOVAL STAGE

In this stage, we coarsely remove the flare from the flare-corrupted image by estimating the coarse flare-corrupted image and the entire flare image. The loss function is formulated as:

$$\mathscr{L} = \begin{cases} \dfrac{1}{N}\sum_{i=1}^{N}|I_{FF}^{i,Coarse} - I_{gt}^{i}| \\[2ex] \alpha\dfrac{1}{N}\sum_{i=1}^{N}|F_{DF}^{i} - F_{gt}^{i}| \end{cases} \tag{1}$$

where $I_{FF}^{i,Coarse}$, $F_{DF}^{i}$, and $F_{gt}^{i}$ represent the $i^{th}$ pixel in the output coarse flare-free image, the output flare image, and the ground truth image.

### 3.2.2 IMAGE REFINEMENT STAGE

After obtaining the flare image which is added on top of the flare-free image and the coarse flare-free image, we introduce the image refinement stage to further remove the flare and artifacts from the coarse flare-free image. We do not adopt an image-to-image translation network as we argue that such a design hardly handles severe degradation. In the image restoration task, simply adopting such a network on image restoration suffers from unpromising results when handling severe degradation (*e.g.*, complicated motion blur, high-level noise). To this end, we accomplish this task from a novel view by extracting the semantic information from the non-polluted area, rewriting the details in the polluted area based on the extracted semantic information, and using multi-scale features extracted from the last stage as the visual prompt. Specifically, we propose the PIN network in this stage

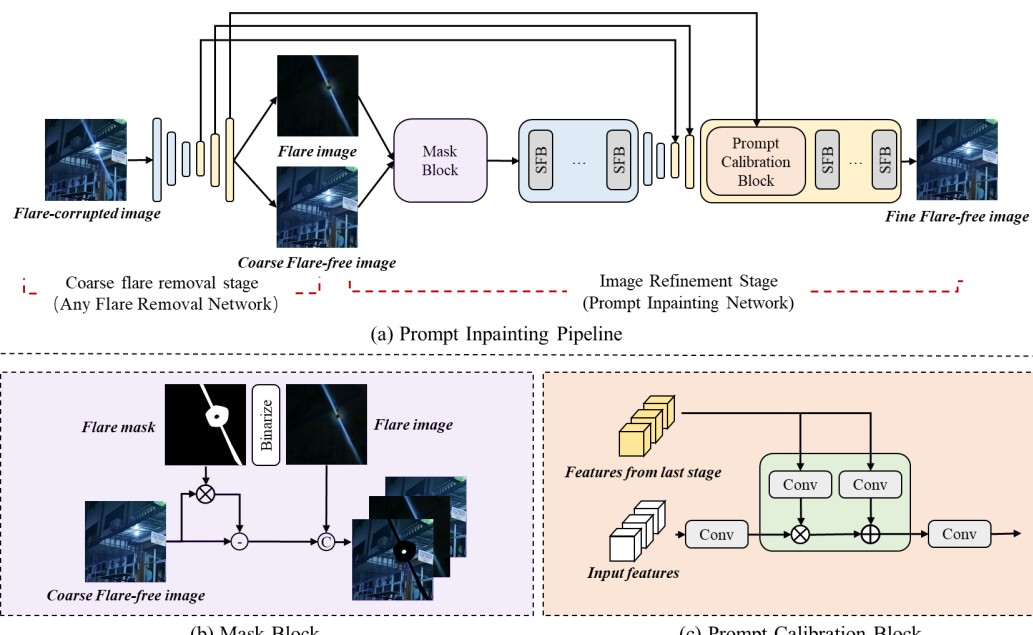

Figure 3: The overview of our PIP pipeline. Our PIP pipeline consists of a coarse flare removal stage and an image refinement stage. The coarse flare removal coarsely estimates the flare and removes the flare from the flare-corrupted image. The image refinement stage further removes the artifacts introduced during the coarse flare removal stage.

which adopts the mask block to generate the input and a U-net neural network to rewrite the missing details.

**The mask block**: The procedure of mask block is depicted in Figure 3 (b). Given the estimated flare image, we first binarize it to obtain the mask with a predefined threshold and multiply it with the coarse flare-free image $I_{FF}^{Coarse}$ to obtain the masked image $I_{FF}^{mask}$. We select the coarse-generated flare-free images instead of the flare-corrupted image as the estimated mask omits the flare with slight luminance and using the flare-corrupted image may leave these flare on the final output. Secondly, we subtract $I_{FF}^{mask}$ with $I_{FF}^{Coarse}$. Finally, we generate the input $I_{FF}^{Input}$ by concatenating the image obtained in the last phase with the estimated flare image $F_{DF}$ and coarse flare-free image $I_{FF}^{Coarse}$. The process is formulated as:

$$Mask = \text{Binarize}(F_{DF} > threshold) \tag{2}$$

$$I_{FF}^{mask} = I_{FF}^{Coarse} \times Mask \tag{3}$$

$$I_{FF}^{mask} = I_{FF}^{Coarse} - I_{FF}^{mask} \tag{4}$$

$$I_{FF}^{Input} = \text{Concat}(I_{FF}^{mask}, F_{DF}, I_{FF}^{Coarse}) \tag{5}$$

**U-shape Structure**: Given $I_{FF}^{Input}$ and features obtained from the last stage $X_{FF}^{coarse}$, we adopt a U-net structure for the inpainting task. Firstly, the encoder in the backbone employs one SFB block Zhang et al. (2023a; 2022) at each level and extracts the multi-scale semantic feature of $I_{FF}^{Input}$. Thereafter, the decoder rewrites the missing details by using $X_{FF}^{coarse}$ as the visual prompt. Concretely, we adopt one prompt calibration block and one SFB block at each level in the decoder and the prompt calibration block is sequentially employed to utilize $X_{FF}^{coarse}$.

**The prompt calibration block**: The prompt calibration block refines the image features by employing the $X_{FF}^{coarse}$ as the visual prompt and the structure is depicted in Figure 3 (c). The formulation of the prompt calibration block is presented as follows:

$$X^i = \text{Conv}(X_{FF}^{i,Coarse}) \times \text{Conv}(X^i) + \text{Conv}(X_{FF}^{i,Coarse}) \tag{6}$$

where $i$ represents the $i^{th}$ level in decoder.

The loss function of this stage is the L1 distance and perceptual loss between the output and the ground truth flare-free image. The formulation is shown as follows:

$$\mathscr{L} = \begin{cases} \dfrac{1}{N} \sum\limits_{i=1}^{N} |I_{FF}^{i,fine} - I_{gt}^i| \\ \mathscr{L}_{per}(I_{FF}^{i,fine}, I_{gt}^i) \end{cases} \tag{7}$$

where $I_{FF}^{i,fine}$ represents the output in refinement stage and $\mathscr{L}_{per}$ means the perceptual loss.

## 4 EXPERIMENTS

### 4.1 DATASETS AND IMPLEMENTATION DETAILS

For the training set, we adopt Flare7K++ Dai et al. (2023a) datasets. Notably, Flare7K and Flare7K++ provide a quantity of flare images, while giving no flare-free images in the training set. We follow the experimental setting in Flare7K which adopts the Flickr24k dataset Zhang et al. (2018a) as flare-free images. To form the data pair for supervised training, we add compound flare images on top of flare-free images to generate nighttime flare-corrupted images. As for flare-free images, we add light source annotations on flare-free images. We use horizontal and vertical flips and channel shuffle to enlarge the training set. And other configurations are consistent with Flare7K++ Dai et al. (2023a). For the testing set, we adopt the real testing set provided by Flare7K++, as the Flare7K++ real set mainly contains scattering and reflective flare. We conduct PIP pipeline on them for ablation experiments and comparison experiments.

Following Dai et al. Dai et al. (2022), we adopt PSNR, SSIM, and LPIPS as our metrics. Specifically, PSNR and SSIM evaluate the average pixel distance and the structure similarity between flare-corrupted images and flare-free images. Furthermore, LPIPS evaluates the feature-level distance between them. We set 4 levels in the PIN network. The channel of the immediate layer is 16. Since SFB Zhang et al. (2023a) has a wider receptive field, we use it as main feature extraction module for PIN. The optimizer for the PIP pipeline adopts the same setting as the Flare7K++ Dai et al. (2023a) and FF-Former Zhang et al. (2023a).

Table 1: Comparison of flare removal methods on Flare7k++ real dataset.

| Model | PSNR ↑ | SSIM ↑ | LPIPS ↓ | G-PSNR ↑ | S-PSNR ↑ |
|---|---|---|---|---|---|
| Zhang et al. Zhang et al. (2020) | 21.022 | 0.784 | 0.1738 | 19.868 | 13.062 |
| Sharma et al. Sharma & Tan (2021) | 20.492 | 0.826 | 0.1115 | 17.790 | 12.648 |
| Wu et al. Wu et al. (2021) | 24.613 | 0.871 | 0.0598 | 21.772 | 16.728 |
| Unet Ronneberger et al. (2015) | 27.189 | 0.894 | 0.0452 | 23.527 | 22.647 |
| HINet Chen et al. (2021) | 27.548 | 0.892 | 0.0464 | 24.081 | 22.907 |
| MPRNet* Zamir et al. (2021) | 27.036 | 0.893 | 0.0481 | 23.490 | 22.267 |
| Restormer* Zamir et al. (2022) | 27.597 | 0.897 | 0.0447 | 23.828 | 22.452 |
| Uformer Wang et al. (2022) | 27.633 | 0.894 | 0.0428 | 23.949 | 22.603 |
| FF-Former Zhang et al. (2023a) | 27.900 | 0.900 | 0.0410 | 24.136 | 23.054 |
| MQANet Li et al. (2025) | 26.600 | 0.890 | 0.0510 | - | - |
| NAFNet Li et al. (2025); Chen et al. (2022) | 25.640 | 0.889 | 0.0600 | - | - |
| SD1.5 Li et al. (2025); Rombach et al. (2022) | 20.920 | 0.764 | 0.1850 | - | - |
| Dai et al. Li et al. (2025); Dai et al. (2023c) | 22.580 | 0.856 | 0.0770 | - | - |
| Difflare Zhou et al. (2024) | 26.063 | 0.898 | - | - | - |
| MFDNet Jiang et al. (2024) | 26.980 | 0.895 | 0.0510 | - | - |
| SFNet Vasluianu et al. (2024) | 27.082 | 0.896 | 0.0460 | 23.640 | 22.013 |
| PromptIR Potlapalli et al. (2023) | 26.843 | 0.893 | 0.0470 | 23.280 | 21.545 |
| Qi et al. Qi et al. (2025) | 28.033 | 0.903 | 0.0420 | 24.537 | 23.614 |
| Sparse-UFormer Wu et al. (2024) | 27.976 | 0.906 | 0.0413 | 24.243 | 23.529 |
| Kotp et al. Kotp & Torki (2024) | 27.662 | 0.897 | 0.0422 | 23.987 | 22.847 |
| SGSFT Ma et al. (2025) | 28.078 | 0.904 | 0.0416 | 24.477 | 23.306 |
| PIP pipeline | | | | | |
| Unet | 28.109 | 0.901 | 0.0412 | 24.592 | 23.539 |
| Uformer | 28.241 | 0.903 | 0.0404 | 24.633 | 23.777 |
| HINet | 28.298 | 0.903 | 0.0406 | 24.964 | 24.203 |
| Restormer* | 28.089 | 0.902 | 0.0427 | 24.768 | 23.432 |
| FF-Former | 28.443 | 0.906 | 0.0392 | 24.837 | 24.111 |

## 4.2 COMPARISON RESULTS

### 4.2.1 RESULTS ON FLARE7K++ DATASETS

To validate the performance of our method on complex scenarios which include both the scattering flare and reflective flare, we show the quantitative results (*e.g.*, PSNR, SSIM, LPIPS) on Table 1 for experiments on Flare7k++ real testing set Dai et al. (2023a) , correspondingly. Furthermore, we also represent the visual comparison results on Figure 4 for experiments on the Flare7k++ real testing set (we employ Uformer Wang et al. (2022) in coarse flare removal network, same as Flare7k++).

From the observation of Table 1, our PIP pipeline equipped with FF-Former Zhang et al. (2023a) has achieved state-of-the-art PSNR and SSIM. Concretely, from the perspective of pixel level, we significantly outperform the state-of-the-art method FF-Former with 0.543dB on PSNR score, 0.006 on SSIM, 0.0018 on LPIPS, 0.701dB on G-PSNR score, and 1.057dB on S-PSNR score. As for visual performance comparison, as shown in Figure 4 and Figure 5, we figure that methods trained in a supervised manner achieve better results and our PIP pipeline equipped with Uformer has essentially eliminated the flare and introduced the least artifacts in most situation. Despite the shape of the flare, round flare with a large radius, flare with a long streak, or the type of the flare, reflective flare, or scattering flare, our method can achieve more realistic and natural results than other baselines in most scenarios.

## 4.3 ABLATION RESULTS

### 4.3.1 IMPACT OF PROMPT INPAINTING NETWORK (PIN)

A systematic evaluation of the impact of PIN's key components on flare removal is presented in Table 2. For fair comparison, we employ Uformer (first row) as the base network in the coarse flare removal stage and test on the flare7k++ real set. In the second row, a simple U-shape refinement network is added to the Uformer, while achieving marginal improvement. It indicates that using a simple U-shape network as a refinement network does not yield

Table 2: Comparison of methods with or without mask block and prompt calibration block (PCB)

| Method | PIN | C | Mask Block | PCB | PSNR |
|--------|-----|---|------------|-----|------|
| | ✗ | - | ✗ | ✗ | 27.63 |
| Uformer | ✓ | 16 | ✗ | ✗ | 27.65 |
| | ✓ | 16 | ✗ | ✓ | 28.06 |
| | ✓ | 16 | ✓ | ✓ | 28.24 |

significant performance gains. Comparing the second row and the third row, we can see that adding the Prompt Calibration Block (PCB) significantly boosts the PSNR from 27.65 to 28.06. This improvement demonstrates that the visual prompt is non-substitutable. The fourth row, with all components of PIN, indicates that using high-quality features to rewrite regions contaminated with flare can achieve better performance.

### 4.3.2 MODEL AGNOSTIC

To demonstrate that our PIP is model-agnostic and can be applied with any other flare removal network, we conduct quantitative and qualitative ablation experiments on Unet Ronneberger et al. (2015), Uformer Wang et al. (2022), and FF-Former Zhang et al. (2023a). The network in the coarse flare removal stage is pretrained for predicting the coarse flare-free image and the flare image on Flare7K++ dataset.

For quantitative comparison, based on Table 1 and Table 3, we can observe that with PIP pipeline, Unet (the channel number of PIP is 16) Ronneberger et al. (2015) gains 0.92dB,

Table 3: The number of parameters, Flops and inference time (512×512, NVIDIA 3090 GPU) with or without PIN.

| Method | PIN | C | Param | Flops | Time | PSNR |
|--------|-----|---|-------|-------|------|------|
| | ✗ | - | 9.0M | 62.4G | 0.0115s | 27.19 |
| Unet | ✓ | 4 | 9.1M | 65.7G | 0.0126s | 27.53 |
| | ✓ | 8 | 10.2M | 70.5G | 0.0126s | 27.73 |
| | ✓ | 16 | 11.8M | 84.9G | 0.0127s | 28.11 |
| | ✗ | - | 20.4M | 162.1G | 0.0723s | 27.63 |
| Uformer | ✓ | 4 | 20.9M | 167.8G | 0.0817s | 27.91 |
| | ✓ | 8 | 21.5M | 175.1G | 0.0818s | 28.05 |
| | ✓ | 16 | 23.6M | 194.3G | 0.0820s | 28.24 |
| | ✗ | - | 46.5M | 407.8G | 0.2109s | 27.90 |
| FF-Former | ✓ | 4 | 47.0M | 413.6G | 0.2146s | 28.26 |
| | ✓ | 8 | 47.6M | 420.8G | 0.2175s | 28.35 |
| | ✓ | 16 | 49.6M | 440.0G | 0.2185s | 28.44 |

0.007, 0.004 on PSNR, SSIM, and LPIPS scores which surpass the state-of-the-art method FF-Former with 0.209dB on PSNR score. Moreover, our PIP pipeline helps Uformer with 0.608dB, 0.009, 0.0024, 0.684dB, and 1.174dB gains on PSNR, SSIM, LPIPS, G-PSNR and S-PSNR scores.

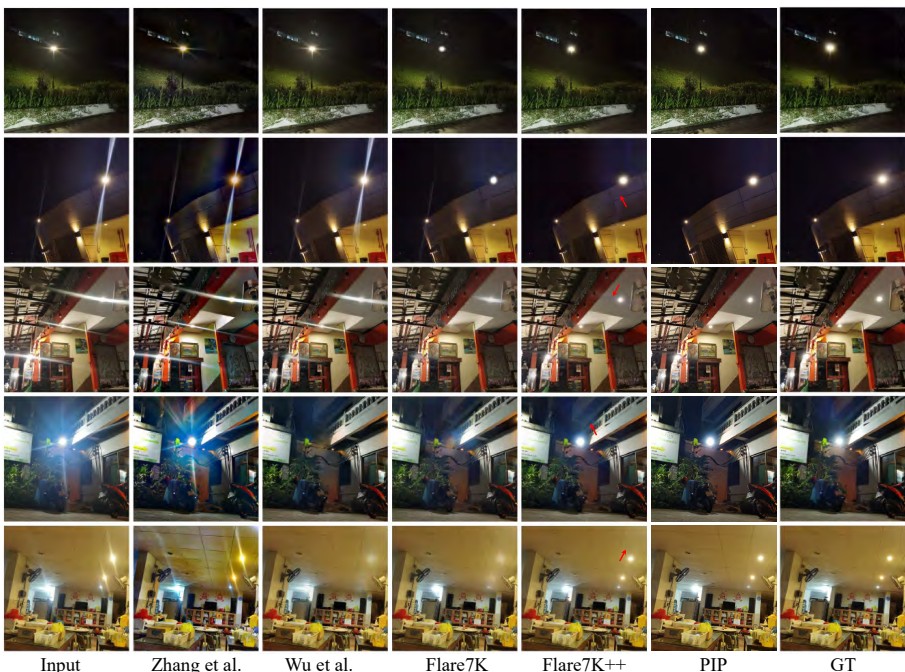

Figure 4: Visual comparison of flare removal on real-world nighttime flare images. Flare7K represents training network with pipeline proposed by Wu et al. Wu et al. (2021). Flare7K++ represents training network with pipeline proposed by Dai et al. Dai et al. (2023a). Our method eliminates the most flare and generates images with the fewest artifacts.

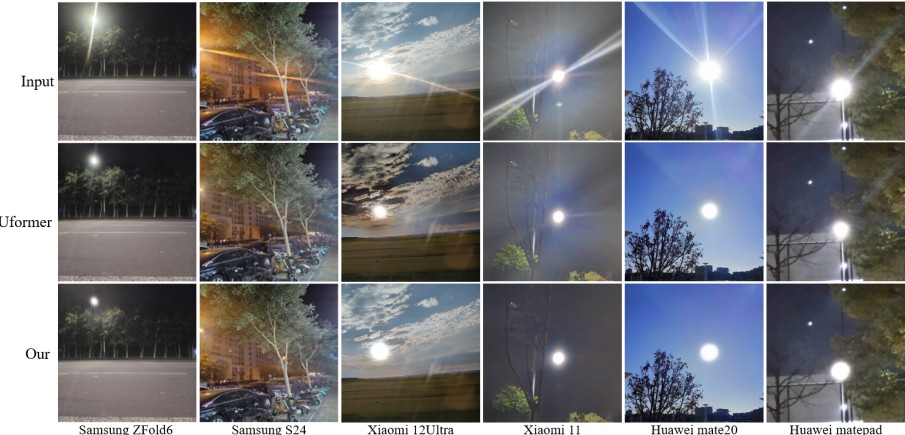

Figure 5: Visual comparison results on real nighttime and daytime flare-corrupted images (without GT). Zoom in for a better view.

As for the state-of-the-art method FF-Former Zhang et al. (2023a), eqipped with our pipeline, it manages to obtain new state-of-the-art results, which achieve 28.443dB, 0.906, 0.0392, 24.837dB and 24.111dB on PSNR, SSIM, LPIPS, G-PSNR and S-PSNR scores and our pipeline provides 0.543dB, 0.006, 0.0018, 0.701dB and 1.057dB promotions with a very small increase in the number of parameters, as shown in Table 3.

As for qualitative comparison, we show the visual comparisons on Figure 6. Compared with the original Unet, Unet + PIP pipeline removes more shimmer within the flare-coorupted images. As for Uformer, more streaks have been eliminated. Previous pipelines hardly identify and eliminate all the flares added to the image and tend to generate severe artifacts when handling strong flares with large overexposed streaks and shimmers. Our PIP pipeline suppresses this phenomenon by rewriting the image details polluted by flare with the extracted semantic information within the non-polluted area and the multi-scale features obtained in the coarse flare removal stage. The experiments demonstrate that our pipeline is model-agnostic.

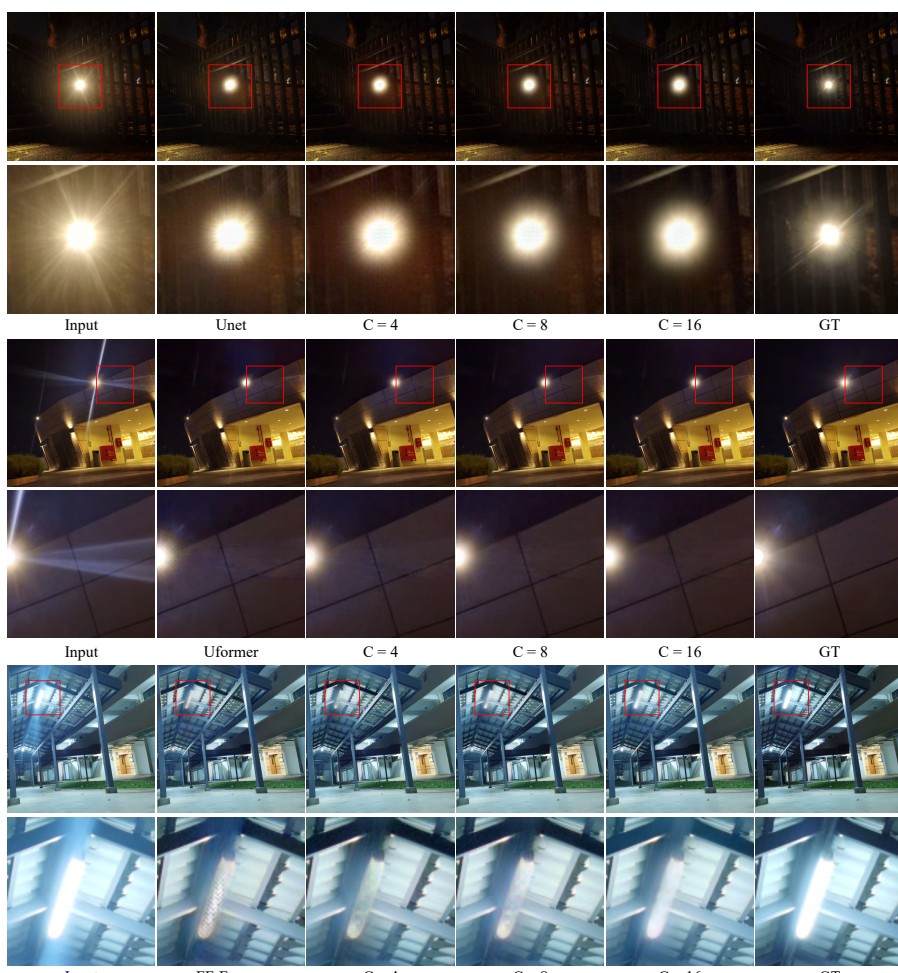

Figure 6: Visual comparison results of ablation experiments on PIP pipeline. Our PIP manages to further suppress artifacts, especially with a weaker network (Unet).

### 4.3.3 NUMBER OF CHANNELS IN PIN

The integration of the Prompt Inpainting Pipeline (PIP) significantly enhances nighttime flare removal across various neural network architectures, as evidenced by the Table 3 and Figure 6. This improvement is attributed to PIP's ability to refine image details corrupted by flare, leveraging multi-scale features from the initial coarse flare removal stage. The data indicates that as the number of channels increases, both objective performance, as measured by PSNR, and subjective visual quality, show a stable enhancement. This trend underscores PIP's effectiveness in leveraging deeper network architectures for improved image restoration, particularly in challenging low-light conditions where flare artifacts are prevalent.

## 5 CONCLUSION

In this paper, we delved into the limitations of existing flare removal methods, which predominantly suffer from the generation of artifacts in restored images. To address this challenge, inspired by in-painting techniques and prompt-based learning, we proposed the Prompt Inpainting Pipeline (PIP). PIP innovatively masks flare-affected regions using the flare image predicted in the first stage and restores these areas with high-quality features from the first-stage decoder act as visual prompts. One of the remarkable features of Prompt Inpainting Network (PIN) of PIP is model-agnostic. It can be seamlessly integrated into any flare removal network without requiring extensive modifi-cations. Comprehensive experimental evaluations across various models and datasets have clearly demonstrated the superiority of PIP.

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

## A ADDITIONAL ABLATION STUDY

### A.0.1 TRAINING PROCESS

The process of our work involves two main steps. Step1: Train flare removal network with Flare7k++ pipline (or download pretrained model like Uformer in Flare7K's github repo). Step2: Frozen flare removal network and train PIN (our training configuration is consistent with Flare7k++ and does not require special design). FF-Former has achieved SOTA performance and SFB Zhang et al. (2023a) has a large receptive field, so we use it as our core module of PIN.

### A.0.2 EFFECTIVENESS AND EFFICIENCY ANALYSIS OF PIP

Based on the results in Figure 7 and the quantitative evidence in Table 3, several insights can be drawn regarding the effectiveness and efficiency of the proposed Prompt Inpainting Network (PIN). First, the left plot of Figure 7 clearly illustrates a consistent upward trend in PSNR as the number of channels $C$ in PIN increases. When $C = 0$, i.e., without employing PIN, the baseline models exhibit comparatively lower PSNR. As channels are gradually increased (e.g., $C = 4, 8, 16$), PSNR improves steadily across different backbones (Unet, Uformer, FF-Former). This trend demonstrates that the inclusion of PIN channels enhances the model's capacity to refine flare-corrupted regions by leveraging multi-scale contextual information from the coarse stage. Importantly, the gains are not marginal: for instance, Table 3 shows that Uformer improves from 27.63 dB (without PIN) to 28.24 dB when PIN with 16 channels is adopted, corresponding to a substantial improvement of 0.61 dB. Similar improvements are observed for Unet (+0.92 dB) and FF-Former (+0.54 dB). These results collectively validate that the visual prompts provided by PIN play a decisive role in guiding the refinement process and reducing artifacts that simple coarse-to-fine pipelines fail to address.

Equally important is the observation from the right plot of Figure 7, which highlights the computational efficiency of PIN. Despite consistent increases in PSNR, the inference time remains nearly unchanged when PIN is integrated. For example, the inference time of Uformer increases only minimally from 0.0723s to 0.0820s even at $C = 16$, as reported in Table 3. A similar pattern is found with Unet and FF-Former, where additional parameters and FLOPs grow modestly but without a significant impact on runtime. This indicates that PIN is computationally lightweight, introducing only a negligible overhead while yielding measurable performance improvements. Such a property is crucial for practical deployment, especially in real-time or resource-constrained environments, where both efficiency and accuracy are indispensable.

Figure 7 and Table 3 jointly demonstrate that PIN effectively balances performance enhancement and computational efficiency. The results confirm that PIN's visual prompt mechanism not only scales with the number of channels to deliver progressive improvements in PSNR but also maintains lightweight inference costs. This dual advantage underscores the robustness and practicality of the proposed approach, affirming its value as a plug-and-play module for flare removal tasks across different backbone architectures.

### A.0.3 RESULTS ON BRACKETFLARE DATASETS

To further validate the robustness of our PIP framework, we conduct experiments on the Bracket-Flare dataset Dai et al. (2023b), which specifically focuses on reflective flare—a particularly challenging scenario due to its symmetry around the optical center and high intensity. Unlike scattering flare, reflective flare often overlaps with important scene structures, making its removal difficult without introducing structural distortions or residual artifacts. As reported in Table 1, our PIP-equipped models consistently surpass strong baselines, including MPRNet, Restormer, and FF-Former. Notably, when integrated with FF-Former, our PIP achieves a PSNR of 49.18 dB, which represents an improvement of 0.77 dB over MPRNet and establishes a new state-of-the-art on this dataset. The SSIM and LPIPS scores further confirm this advantage, showing that our method not only restores images with higher pixel-level fidelity but also achieves better perceptual quality.

Visual results in Figure 8 corroborate these quantitative findings. Competing methods often leave faint residual flare streaks or introduce excessive smoothing in flare regions, leading to the loss of fine structures around light sources. In contrast, our approach effectively suppresses reflective flare while preserving structural details and texture consistency, producing results that are visually closer to the ground truth. The superior performance can be attributed to the prompt-guided inpainting

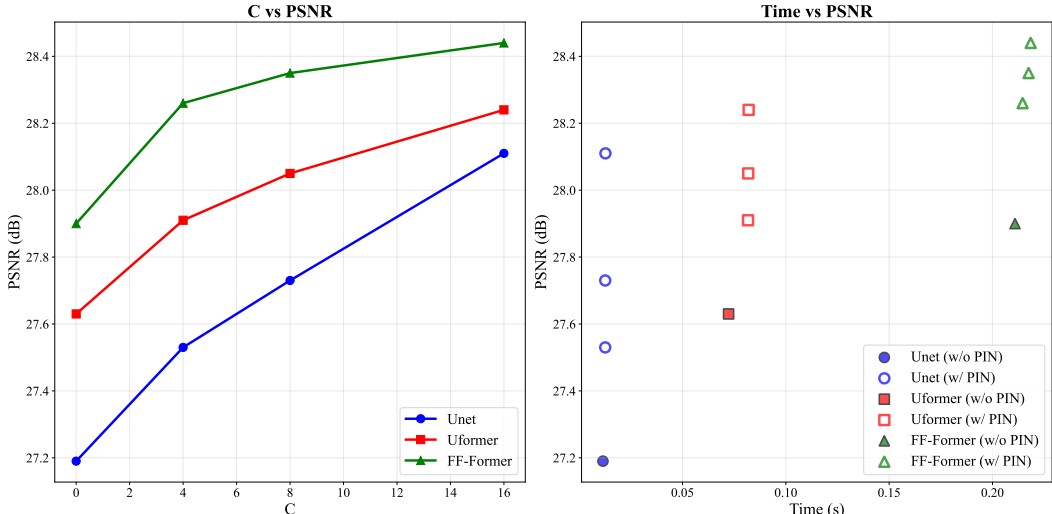

Figure 7: Relationship between the number of PIN channels and PSNR (left), and inference time versus PSNR (right), demonstrating both performance gains and lightweight efficiency of PIN.

mechanism of PIN, which leverages non-polluted regions and multi-scale semantic features from the coarse stage to reconstruct corrupted areas more faithfully.

These results highlight the generalization ability of our PIP framework beyond the Flare7K++ dataset. By successfully handling both scattering and reflective flare, PIP demonstrates its adaptability to diverse real-world flare phenomena. This reinforces its potential as a model-agnostic and practically deployable solution for nighttime imaging applications where various flare types coexist.

Table 4: Comparison of flare removal methods on BracKetFlare dataset. The highest score has been shown on **bold**.

| Method | BracKetflare | | |
|---|---|---|---|
| | PSNR ↑ | SSIM ↑ | LPIPS ↓ |
| Wu et al. Wu et al. (2021) | 26.13 | 0.895 | 0.055 |
| Unet Ronneberger et al. (2015) | 47.00 | 0.897 | 0.006 |
| HINet Chen et al. (2021) | 48.03 | 0.994 | 0.003 |
| MPRNet* Zamir et al. (2021) | 48.41 | 0.994 | 0.004 |
| Restormer* Zamir et al. (2022) | 48.11 | 0.994 | 0.004 |
| Uformer Wang et al. (2022) | 47.47 | 0.991 | 0.003 |
| FF-Former Zhang et al. (2023a) | 47.98 | 0.992 | 0.003 |
| Zhang et al.Zhang et al. (2018b) | - | 0.830 | 0.074 |
| Dong et al.Dong et al. (2021) | - | 0.907 | 0.041 |
| Soomin Kim et al.Kim et al. (2020) | - | 0.857 | 0.069 |
| He et al.He et al. (2025) | - | 0.967 | 0.038 |
| Unet + PIP (Ours) | 48.50 | 0.992 | 0.004 |
| Uformer + PIP (Ours) | 48.69 | 0.994 | 0.002 |
| FF-former + PIP (Ours) | **49.18** | **0.994** | **0.002** |

A.0.4 COMPARISON WITH INPAINTING METHODS

To further test whether other inpainting method can be useful for the flare removal task, we also compare it with ZITS Dong et al. (2022) method and adopt the official code for inference, and SDXL Podell et al. (2023) inpainting method and adopt the demo from huggingface[1]. The visual performance of the comparison is shown in Figure 9. Severe artifacts happen in the image generated by ZITS as the mask area is too complicated and uncertain to recover when much information has been discarded by the simple mask operation and no visual prompt can be used. We conducted 4

---

[1]https://huggingface.co/spaces/diffusers/stable-diffusion-xl-inpainting

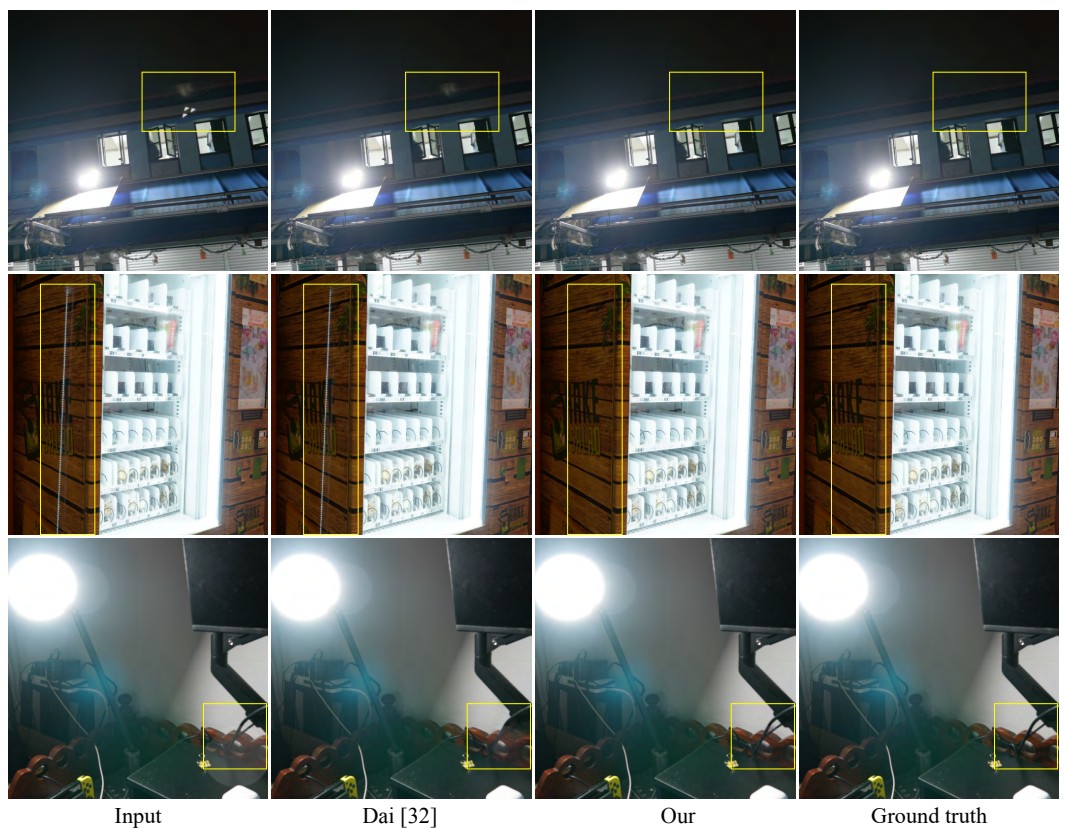

| Input | Dai [32] | Our | Ground truth |

Figure 8: Visual comparison of BraceKet flare dataset. Zoom in for a better view. We mark the area around the reflective flare for a better look.

iterations of flare reduction using the SDXL inpainting algorithm. However, as depicted in Figure 9, it is evident that SDXL not only fails to completely eliminate flare but also introduces new artifacts, manifesting as additional objects and varying styles of flare generation.

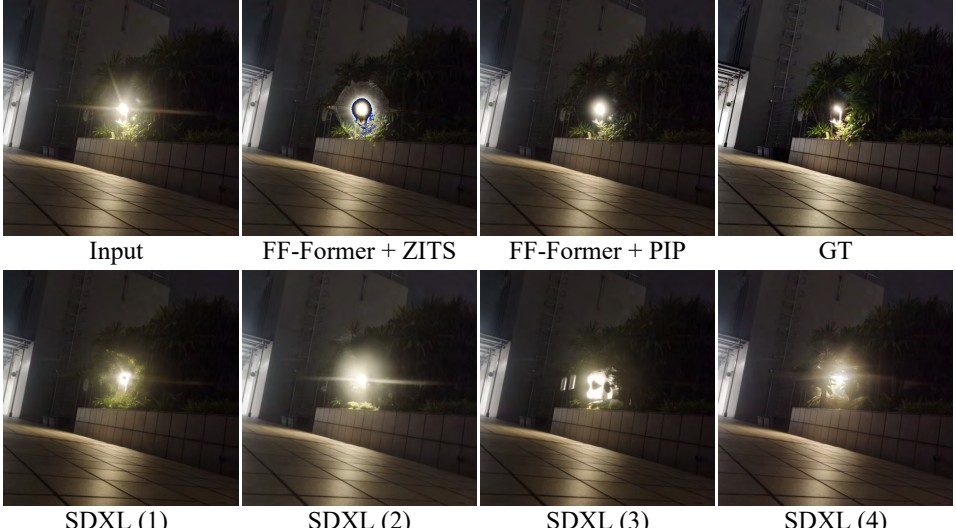

Figure 9: Visual comparison results on PIP pipeline VS inpainting network ZITS Dong et al. (2022) and SDXL Podell et al. (2023) inpainting method. Severe artifacts happen in the image generated by FF-Former + ZITS, SDXL generates new objects while removing flare, which not only cannot completely eliminate flare, but also generates other styles of flare.

### A.0.5   DISCUSSION

To suppress the artifacts created during the flare removal process, we propose the two-stage pipeline named PIP. The PIP pipeline borrows the idea from the inpainting task and adopts a mask block to construct the input of the inpainting neural network. However, as shown in Figure 9, simply adopting the inpainting method will lead to essential content loss as plenty of the image content is hidden by the flare, which leads the current inpainting method hardly recovers them and the extracted flare mask is error-prone, which makes the inpainting method mistakenly rewrites the flare instead of image content. Therefore, we adjust the inpainting method for the flare removal task. By adopting the features extracted from the coarse flare removal stage as inference, the prompt calibration block in the image refinement stage manages to rewrite the blocked image content.

## B   LIMITATIONS AND FUTURE WORK

In this paper, we introduce the Prompt Inpainting Pipeline (PIP), which achieves state-of-the-art performance in nighttime flare removal by leveraging a model-agnostic two-stage framework. While our core contribution lies in redefining the coarse-to-fine paradigm through the integration of visual prompts and inpainting concepts, the current design of the Prompt Inpainting Network (PIN) does not include explicit architectural optimizations tailored to the unique characteristics of flare patterns, such as their spatial distribution, spectral properties, or geometric structures. And the current implementation adopts generic restoration backbones, which may not fully exploit domain-specific priors inherent to flare corruption. For future work, developing specialized neural network components that explicitly model the physical properties and visual signatures of different flare types has the potential to further enhance the pipeline's ability to handle complex flare scenarios and achieve even more refined restoration results.

## C   ADDITIONAL VISUAL RESULTS

In this part, we provide additional visual results compared to the Flare7k++Dai et al. (2023a) pipeline with Uformer Wang et al. (2022).

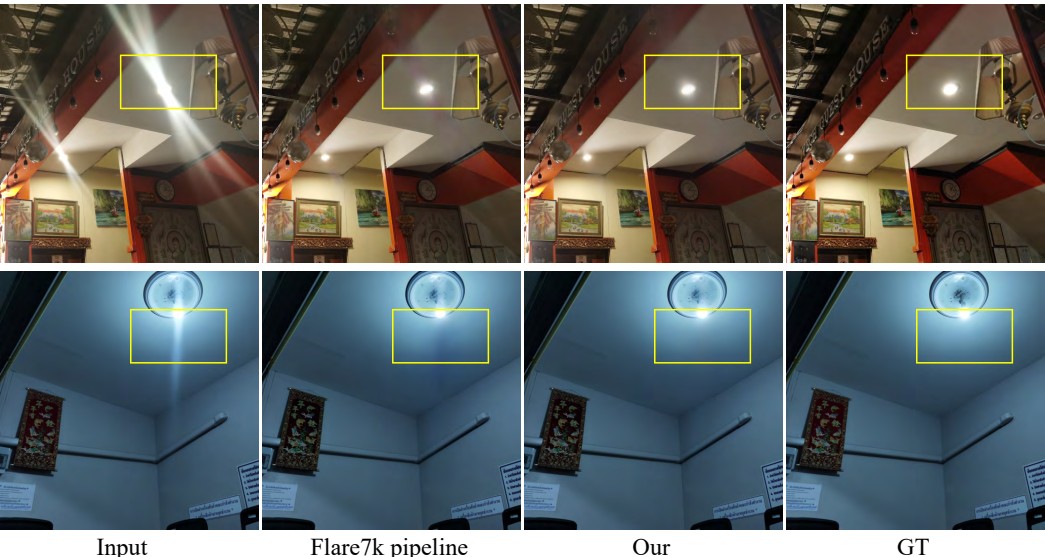

| Input | Flare7k pipeline | Our | GT |

Figure 10: Visual results achieved by different methods on the Flare7k real dataset.

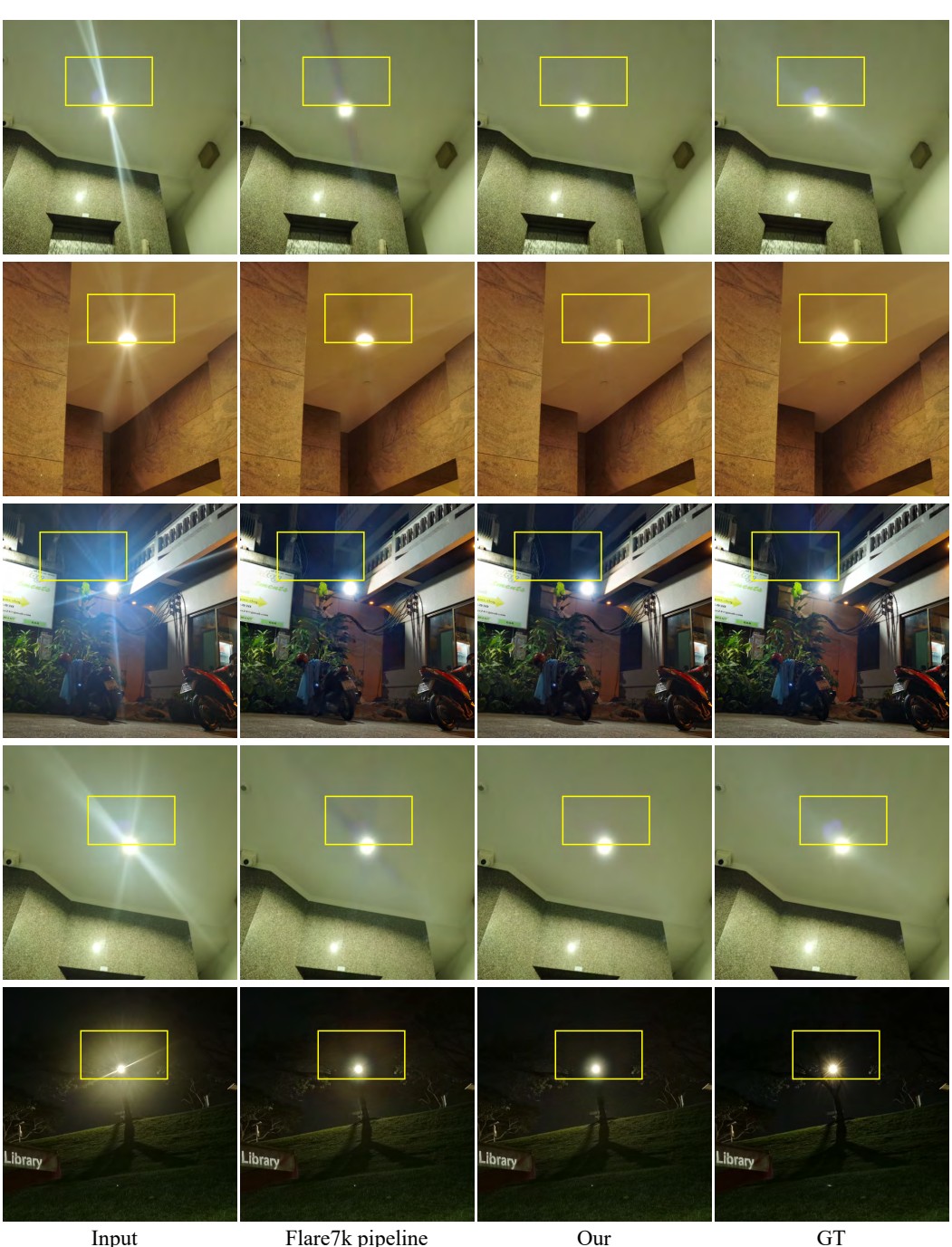

|       |                 |     |    |
|-------|-----------------|-----|----|
| Input | Flare7k pipeline | Our | GT |

Figure 11: Visual results achieved by different methods on the Flare7k real dataset.

