# OpenReview forum: "Revisiting Coarse-to-fine Paradigm in Nighttime Flare Removal via Visual Prompt"
_ICLR.cc/2026/Conference — ICLR 2026 Conference Withdrawn Submission_

### Official Review · Reviewer_2WPf · 2025-10-23

**Soundness:** 2
**Presentation:** 3
**Contribution:** 2
**Rating:** 2
**Confidence:** 5

**Summary:**

This paper proposes a plug-and-play pipeline, namely PIP, for nighttime flare removal. This pipeline employs a coarse-to-fine manner, which consists of a coarse flare removal stage and an image refinement strategy. A network, PIN, is developed in the image refinement strategy, where a prompt calibration block is designed to guide the inpainting process. Experiments demonstrate the effectiveness of the proposed method.

**Strengths:**

The proposed PIP addresses the artifacts in the results of flare removal, especially in the streaky or shimmering regions. The pipeline is model-agnostic and can be integrated with existing U-Net-based networks. Experiment results demonstrate its effectiveness.

**Weaknesses:**

1. The core idea of the proposed pipeline is to employ an inpainting stage after flare removal. It utilizes the flare mask as guidance for inpainting. However, such an idea is not novel, since similar ideas have been explored in other restoration tasks, such as [1][2]. A clarification of novelty is encouraged.
2. The effectiveness of PIN is not clear. Though a comparison between PIN and other inpainting methods is conducted in the appendix, the compared methods (ZITS, SDXL) use the official checkpoints and are not finetuned in the task of flare removal. Since PIN is tuned in flare removal, the comparison may not be fair.
3. The visual results may not be convincing. In the example from ``Samsung S24" of Figure 5, though Uformer yields more shimmer, it removes more glare around the light.

[1] Style-Guided Shadow Removal, ECCV, 2022.

[2] Unsupervised Night Image Enhancement: When Layer Decomposition Meets Light-Eﬀects Suppression, ECCV, 2022.

**Questions:**

1. The flare removal method often requires a light source blend process, since it is hard to distinguish between flare and light source. This paper does not introduce this crucial process. Can these details be provided?

---

### Official Review · Reviewer_LqTk · 2025-10-31

**Soundness:** 3
**Presentation:** 4
**Contribution:** 3
**Rating:** 6
**Confidence:** 3

**Summary:**

This paper tackles limitations in nighttime flare removal—artifacts in restored images and ineffective traditional coarse-to-fine methods. Inspired by inpainting and prompt learning, it proposes the model-agnostic Prompt Inpainting Pipeline (PIP): a coarse flare removal stage, plus a refinement stage with a Prompt Inpainting Network (PIN) that uses coarse-stage flare masks (to exclude corrupted pixels) and decoder features (as visual prompts for structural consistency). Tested on Flare7k++ and BracketFlare via PSNR/SSIM/LPIPS, PIP integrates seamlessly with models like Unet/Uformer/FF-Former, cuts artifacts, achieves SOTA (e.g., FF-Former+PIP: 28.443dB PSNR on Flare7k++, 49.18dB on BracketFlare) with minimal overhead (≥1% params), and notes future work on flare-specific architecture optimization.

**Strengths:**

1. It reinvents the coarse-to-fine paradigm for flare removal by integrating inpainting (flare mask to isolate corrupted pixels) and prompt learning (coarse-stage features as visual prompts)—a novel, task-specific fusion that fixes prior method flaws, rather than inventing entirely new concepts.
2.Rigorous in design (model-agnostic, minimal overhead: ≥1% params) and validation: comprehensive tests on Flare7k++/BracketFlare, clear technical details (e.g., mask equations), ablation studies proving core mechanisms, and SOTA results (e.g., FF-Former+PIP hits 28.443dB PSNR on Flare7k++).
3. Follows a logical problem-solution flow, defines jargon upfront (e.g., "model-agnostic"), and links figures/captions tightly to text—making complex ideas accessible even to non-specialists.
4. Practically, it’s deployable (retrofittable, real-time efficient) for real-world imaging (mobile, autonomy). Academically, it rescues coarse-to-fine for low-level vision, bridges subfields (prompt learning + restoration), and sets a new benchmark.

**Weaknesses:**

1.Narrow Flare Type Testing: Only tests scattering/reflective flare (Flare7k++/BracketFlare) but ignores multi-source, dynamic, or asymmetrical reflective flare. Fix: Add tests on RealFlare dataset (multi-source) and a custom asymmetrical Flare7k++ subset; analyze failure modes (e.g., mask overlap) to adjust mechanisms.
2.Shallow "Model Agnosticism": Only validates on U-shape architectures (Unet/Uformer), not non-U-shape (NAFNet) or lightweight edge models (MobileUnet). Fix: Test PIP on NAFNet (adapt prompts to its feedforward features) and MobileUnet; report mobile GPU efficiency (e.g., Adreno 650) and optimize PIN channel count (C=2/1) for edge use.
3.Unclear Visual Prompt Value: No comparison to basic feature reuse (e.g., Zamir et al. 2021) or interpretability. Fix: Add an ablation with simple feature concatenation (no prompt block) to isolate prompt impact; use attention maps to show how prompts guide refinement.

**Questions:**

1.Author focus on scattering and reflective flare (Flare7k++/BracketFlare), but real-world nighttime imaging often includes multi-source flare (e.g., overlapping streetlights) and asymmetrical reflective flare (e.g., off-center light sources breaking BracketFlare’s "optical center symmetry" prior). Has PIP been tested on these scenarios? If not, do you anticipate bottlenecks (e.g., mask block failing to distinguish overlapping flare regions, visual prompts becoming noisy with conflicting context), and what modifications (e.g., adaptive mask thresholding, multi-scale prompt calibration) might be needed to handle them?
2. Author claim PIP is "model-agnostic" but only validate it on U-shape architectures (Unet, Uformer, FF-Former). For non-U-shape backbones (e.g., NAFNet, a feedforward model without a traditional decoder) or extremely lightweight edge models (e.g., MobileUnet, <2M parameters), how do you adapt PIP’s "coarse-stage decoder features as visual prompts" (since NAFNet lacks a decoder)? For lightweight models, what is the parameter ratio of PIN to the base model (e.g., PIN params / MobileUnet params), and does PIP’s overhead remain "negligible" for edge deployment?
3.The frame "visual prompts from coarse-stage features" as a core innovation but do not distinguish it from standard feature reuse (e.g., skip connections in MPRNet, Zamir et al. 2021). Have you run an ablation where the prompt calibration block is replaced with simple feature concatenation (coarse features + refinement input, no gating as in Eq. 6)? What performance gap exists between this baseline and PIP’s prompt design?
4.Can you provide interpretability results (e.g., attention maps, feature visualization) showing how visual prompts guide refinement (e.g., prioritizing edges over texture to reduce artifacts)?
5.Efficiency analysis uses an NVIDIA 3090 (desktop GPU), but PIP’s target use cases (e.g., mobile photography) rely on edge GPUs (e.g., Qualcomm Adreno 650). Please add inference time benchmarks on a mobile GPU for lightweight backbones (e.g., Unet+PIP, MobileUnet+PIP); if overhead is prohibitive, test a "lightweight PIN" variant (e.g., C=2 or C=1 channels instead of C=16) and report whether performance gains are retained.
6.Results show PIP works well for moderate flare, but how does it perform on extreme flare cases (e.g., large overexposed regions blocking >50% of the image, or flare overlapping with high-detail areas like text/signs)? Do artifacts reemerge in these cases, and if so, is the bottleneck the mask block (failing to isolate all corrupted pixels) or the prompt mechanism (insufficient context from the coarse stage)?

---

### Official Review · Reviewer_fJS5 · 2025-10-31

**Soundness:** 2
**Presentation:** 2
**Contribution:** 2
**Rating:** 4
**Confidence:** 4

**Summary:**

This paper addresses the problem of image flare removal. The authors propose a two-stage framework called the Prompt Inpainting Pipeline (PIP). The pipeline first performs coarse flare removal and predicts a flare mask, then uses a refinement stage called the Prompt Inpainting Network (PIN) that leverages decoder features from the coarse stage as visual prompts to guide structural restoration. The method is presented as a model-agnostic, plug-and-play enhancement applicable to existing flare removal architectures. Experiments on Flare7K++ demonstrate improved quantitative and qualitative results, with minimal parameter overhead.

**Strengths:**

1. Interesting conceptual reframing: The idea of using prompt-guided inpainting for flare removal is conceptually interesting and bridges prompting concepts with low-level image restoration.
2. Plug-and-play modularity: The proposed pipeline can be integrated into arbitrary flare removal architectures with minimal modification, making it practically useful for applied restoration research.
3. Improved qualitative realism and cross-model effectiveness: The proposed PIP framework enhances the performance of diverse backbone architectures, including U-Net, UFormer, and FF-Former. Also, visual results on real-world flare images demonstrate cleaner recovery of specular regions and fewer color artifacts. showing that the approach generalizes well across different model families and flare patterns.
4. Clarity and readability: The paper is generally well-written and easy to follow, with a clear description of the two-stage pipeline and its motivation.

**Weaknesses:**

1. Incremental improvement: Despite the appealing terminology, the technical novelty is limited. The approach largely reuses existing architectures with a feature-conditioning mechanism that resembles conventional refinement or modulation blocks rather than a fundamentally new learning strategy.
2. Ambiguity in the claimed prompting mechanism: The “prompt” concept here functions more as feature-level refinement or feature modulation than as genuine prompt-based learning. The naming may therefore be somewhat misleading compared to established prompt-driven paradigms such as PromptIR (NeurIPS'23), which provides a more general and theoretically grounded treatment of similar ideas.
3. Task-specific focus: Although the framework is described as model-agnostic, its core mechanism—relying on predicted flare masks and light-source information—is tightly coupled to the flare removal setting and datasets. This dependence limits the framework’s applicability to other degradation types such as rain streaks, lens blur, or motion blur.
4. Need for deeper analysis: The paper lacks detailed ablations or visualizations analyzing what the prompts encode, how they affect feature propagation, and why they improve artifact suppression. Also, There’s no deep analysis or theoretical insight into why prompt-based inpainting helps beyond empirical results.
5. Limited evaluation and generalization evidence: The experiments are confined to the Flare7K++ dataset. To substantiate claims of model-agnostic design and broad applicability, the paper should include evaluations on other datasets, and demonstrate robustness under different lighting, sensor, and noise settings.

**Questions:**

1.Clarification on “prompt” mechanism:
Can you clarify in more detail how your “visual prompts” differ from conventional feature modulation or feature modulation? Specifically, what makes this prompting mechanism conceptually aligned with prompt-based learning rather than standard feature reuse?
2.Applicability to other flare datasets:
The proposed PIP framework depends on the availability of light source annotations to generate the flare mask in the coarse stage. How would the method perform on flare datasets that lack such annotations or masks (e.g., datasets other than Flare7K++)? Can the pipeline operate without explicit light source supervision, or is this dependency fundamental to its design?
3.Generalization to other degradations: Given that the proposed PIP framework relies on a predicted flare mask derived from light source information, its current formulation appears tightly coupled to the flare removal task. Could the authors clarify whether the same coarse-to-fine prompting principle could be extended to other degradation types (e.g., rain streaks, lens blur, dirt), and if so, how the notion of “corruption mask” and “visual prompts” would be adapted in those cases? Demonstrating or discussing this would strengthen the claim of model-agnostic generality.

---

### Official Review · Reviewer_XETc · 2025-11-02

**Soundness:** 2
**Presentation:** 2
**Contribution:** 2
**Rating:** 2
**Confidence:** 4

**Summary:**

This paper addresses the problems of artifacts in existing methods and the ineffectiveness of traditional coarse-to-fine strategies in nighttime image flare removal, proposing a model-agnostic Prompt Inpainting Pipeline (PIP). Innovatively integrating inpainting ideas with visual prompts, PIP consists of two stages: a coarse flare removal stage that initially eliminates flare and generates a flare mask, and an image refinement stage that uses a Prompt Inpainting Network (PIN) to isolate corrupted regions via the mask while guiding detail restoration with features from the coarse stage as visual prompts. The core contributions are: redefining the coarse-to-fine process for flare removal to overcome the limitations of traditional strategies; designing a model-agnostic PIN module that can be seamlessly integrated into various existing architectures; and extensive experiments verifying that PIP significantly improves restoration quality on multiple datasets, reducing artifacts while maintaining computational efficiency. This method provides a new effective approach for flare removal with strong practical application value.

**Strengths:**

1. The experimental design is comprehensive, covering various scenarios such as scattering flare and reflective flare. Comparisons with multiple SOTA models on datasets like Flare7K++ and BracketFlare, along with quantitative and visual results, fully support the effectiveness of the method. Ablation experiments detailedly verify the necessity of core components such as the PIN module, mask block, and prompt calibration block, with complete analysis of parameter increment and computational efficiency.
2. The paper has a well-structured hierarchy, with coherent logic from problem formulation, related work to method design and experimental verification. Framework diagrams and comparison charts intuitively show the workflow and restoration advantages of PIP. Technical details are elaborated in detail, and the literature review is comprehensive, facilitating readers' understanding of the research background and method innovations.
3. The PIN module is model-agnostic, which can be seamlessly integrated into various existing flare removal networks based on CNN or Transformer without large-scale modifications. It only introduces minimal computational overhead (minimum 1% parameter increment), balancing performance and efficiency, and adapting to practical deployment needs.

**Weaknesses:**

1. Single mask generation strategy: The mask is only generated by binarizing the flare image from the coarse stage with a fixed threshold, without considering the continuity of flare intensity and fuzzy boundary scenarios. This may lead to inaccurate masks, thereby affecting the refinement effect; the performance of adaptive threshold or dynamic mask generation strategies is not verified.
2. Lack of flexibility in visual prompt design: In the refinement stage, only the decoder features from the coarse stage are used as visual prompts, and the effect of different levels and types of features (such as encoder features, multi-scale fused features) as prompts is not explored. The feature fusion method of the prompt calibration block is relatively simple, without considering the differentiated adaptation of flare types and intensities.
3. Insufficient verification of extreme scenarios: Experiments are mainly focused on common scattering flare and reflective flare, and the performance in extreme scenarios such as strong flare, complex backgrounds (e.g., dense light sources, texture-rich scenes), and coexistence of flare and noise is not fully verified; the adaptability of the method to images of different resolutions is not evaluated.
4. Insufficient originality depth, failing to meet bar of ICLR: The coarse-to-fine paradigm is highly prevalent in the field of image restoration (e.g., MPRNet for deblurring, Restormer for deraining all adopt similar ideas). Although the paper combines it with visual prompts and inpainting to adapt to flare removal, it is essentially a modification of a mature framework rather than a groundbreaking innovation. Compared with flare removal-related works published in CVPR in recent years (such as new paradigms driven by physical models, and brand-new architectures for cross-modal fusion), this method lags in the originality of core ideas and struggles to meet the top conference's requirement for "field breakthroughs".

**Questions:**

1. What is the basis for using a fixed threshold for mask generation? Have adaptive thresholds (e.g., based on image brightness distribution, flare intensity statistics) or dynamic mask generation methods been tried? How does different threshold settings affect the final restoration effect?
2. Why are the coarse decoder features selected as visual prompts in the refinement stage? Will the performance differ if encoder features, multi-scale fused features, or cross-stage features are used? Can the feature fusion weights of the prompt calibration block be adaptively adjusted to fit different flare scenarios?
3. How does the method perform in extreme scenarios such as strong flare (e.g., large-area overexposed streaks), complex backgrounds (e.g., dense light sources in urban night scenes), and coexistence of flare and noise? Are there corresponding experimental verifications or optimization strategies?
4. The paper only states that the parameter increment after PIN integration is at least 1%, but fails to clarify which components (e.g., mask block convolution layers, prompt calibration block weights, U-shaped structure new channels) PIN’s trainable parameters correspond to in different baseline models (Unet, Uformer, FF-Former). Could you explain the component attribution of PIN’s trainable parameters in each baseline model?

---

### Note · Authors · 2025-11-12

I have read and agree with the venue's withdrawal policy on behalf of myself and my co-authors.